# Comparison Study of the Physicochemical Properties, Amino Acids, and Volatile Metabolites of Guangdong Hakka *Huangjiu*

**DOI:** 10.3390/foods12152915

**Published:** 2023-07-31

**Authors:** Min Qian, Fengxi Ruan, Wenhong Zhao, Hao Dong, Weidong Bai, Xiangluan Li, Xiaoyan Liu, Yanxin Li

**Affiliations:** 1College of Light Industry and Food Sciences, Academy of Contemporary Agricultural Engineering Innovations, Zhongkai University of Agriculture and Engineering, Guangzhou 510225, China; qianmin@zhku.edu.cn (M.Q.); rfx13632272705@163.com (F.R.); weidong_bai2010@163.com (W.B.); leexiangluan@163.com (X.L.); lxyan0344@163.com (X.L.); liyanxin071243@163.com (Y.L.); 2Guangdong Provincial Key Laboratory of Lingnan Specialty Food Science and Technology, Zhongkai University of Agriculture and Engineering, Guangzhou 510225, China; 3Key Laboratory of Green Processing and Intelligent Manufacturing of Lingnan Specialty Food, Ministry of Agriculture, Guangzhou 510225, China

**Keywords:** Guangdong Hakka *Huangjiu*, physicochemical properties, amino acids, organic acid, volatile metabolites

## Abstract

The physicochemical properties, amino acids, and volatile metabolites of 20 types of Guangdong Hakka *Huangjiu* were systematically compared in this study. Lower sugar contents were detected in LPSH, ZJHL-1, and GDSY-1, but the total sugar contents of the other types of Guangdong Hakka *Huangjiu* were more than 100 g/L (which belonged to the sweet type). Among them, a lower alcohol content was found in GDSY-1 (8.36 %vol). There was a significant difference in the organic acid and amino acid composition among the 20 Guangdong Hakka *Huangjiu* samples, especially the amino acid composition. However, bitter amino acids as the major amino acids accounted for more than 50% of the total amino acids. A substantial variation in volatile profiles was also observed among all types of Guangzhou Hakka *Huangjiu*. Interestingly, MZSK-1 had different volatile profiles from other Guangzhou Hakka *Huangjiu* samples. According to gas chromatography olfactometry (GC-O), most of the aroma-active ingredients identified in Guangdong Hakka *Huangjiu* were endowed with a pleasant aroma of “fruity”.

## 1. Introduction

Chinese rice wine, known as *Huangjiu* or yellow wine, is the oldest form of wine in the world and has been highly favored by consumers in China for more than five thousand years. According to a previous report, approximately 200 million liters are consumed in China every year, and their sales amount has ranged from RMB 8.96 billion to RMB 19.83 billion [1]. *Huangjiu* is fermented from cooked glutinous rice or millet with Qu (including Hong Qu, Wheat Qu, and Jiu Qu) or yeast as a starter. The technology for making rice wine and the microorganisms used in the starter cultures have been widely reported [2,3,4,5]. *Huangjiu* preparation is commonly carried out at 28 °C for 5–7 days (primary fermentation) and 15–20 °C for 20–25 days (secondary fermentation) in a natural open fermentation environment, which directly impacts the resulting wines’ quality. The protein and starch in rice can be decomposed into amino acids, peptides, and low-molecular low-molecular-weight sugars during *Huangjiu* fermentation process, and a range of minor volatile flavor metabolites are also generated [6]. Some studies exhibited that *Huangjiu* has a series of physiological functions including scavenging free radicals, resisting aging, and relieving cardiovascular diseases that are strongly associated with the increases in active component contents (including glutathione and polyphenols) [7]. In addition, the open fermentation environment leads to a significant difference in the quality of wine from different regions. It is well known that *Huangjiu* can be divided into four types on the basis of the geographical situation, namely Shaoxing rice wine, Jimo old wine, Shanghai old wine, and Hakka *Huangjiu*. Among them, Hakka *Huangjiu*, known as glutinous rice wine, is a famous alcoholic beverage in South China that has a history of more than 1000 years [8]. At present, more than 15 kinds of Guangdong Hakka *Huangjiu* are consumed in South China. However, the differences in the physicochemical properties, amino acids, and volatile metabolites of Hakka *Huangjiu* in the market has been rarely reported.

A mixed fermentation starter mainly containing Hong Qu, Wheat Qu, and Yao Qu is often used in the process of Hakka *Huangjiu* preparation that can elevate the raw materials’ utilization rate, suppress the increase in total acid content, and increase the contents of amino acids and total sugars [8]. The starter is commonly inoculated with a variety of bacteria and fungi. Numerous pigments, enzymes, and metabolites secreted from these microorganisms can promote the formation of a unique flavor [9]. In addition, proper consumption of *Huangjiu* can reduce the risk of some diseases such as heart disease, hypertension, and insomnia due to its relative high contents of amino acids, organic acids, vitamins, and trace elements [10]. However, the alteration in nutrient substance contents is mainly affected by the compositions of microorganisms in the starter. A previous study showed that Hakka *Huangjiu* has low ethanol contents (about 14–17% by volume) and medium total sugar contents (about 15–100 g/L) [11]. Hakka *Huangjiu* can be divided into four types on the basis of total sugar contents, namely the dry type (less than 15 g/L), semi-dry type (15–40 g/L), semi-sweet type (40–100 g/L), and sweet type (more than 100 g/L) [12]. Therefore, it is necessary to provide a data comparison of the total sugar contents in various types of Hakka *Huangjiu* to assist in consumers’ choices based on the demand for sugar.

Therefore, the aim of the present study was to investigate and compare the physicochemical properties and volatile metabolites of 20 types of Guangdong Hakka *Huangjiu,* including the dry type, semi-dry type, semi-sweet type, and sweet type by means of high-performance liquid chromatography (HPLC), headspace solid-phase micro-extraction coupled with gas chromatography–mass spectrometry (HS-SPME-GC-MS), and gas chromatography–olfactometry (GC-O) analysis. The results obtained may provide a better understanding of Hakka *Huangjiu* quality, which may be helpful in the choices by consumers and in the preparation of standards.

## 2. Materials and Methods

### 2.1. Chemicals and Reagents

Twenty types of Guangdong Hakka *Huangjiu* were obtained from the local market (Yonghui supermarket, Guangzhou, China). Organic acids including oxalic acid, malic acid, acetic acid, lactic acid, tartaric acid, succinic acid, pyruvic acid, and citric acid were purchased from Yuanye Biotechnology Co., Ltd. (Shanghai, China). Standards of seventeen amino acids were purchased from the Sigma-Aldrich company (Shanghai, China). Fructose, glucose, sucrose, maltose, and lactose were purchased from the Chemical Reagent Factory (Guangzhou, China).

### 2.2. Physicochemical Analysis

The total sugar content, alcohol concentration, major simple sugars, organic acids, and amino acids were detected according to the methods in our previous studies [5,8] and another report [13].

### 2.3. Volatile Compounds Analysis

Volatile compounds of samples were measured using HS-SPME-GC-MS on the basis of our previous report [5].

### 2.4. GC-O Analysis

The volatiles were further screened using GC-MS coupled with an olfactory detection port. The program conditions of GC-O-MS were according to a previous report [14]. The volatile profiles extracted were separated with a split at a ratio of 1:2. The column gradient program was as follows: oven program started at 40 °C for 3 min and then elevated to 250 °C at the rate of 5 °C/min and kept at 250 °C for 12 min. Following the methodology described by Xu et al. [15], eight panelists consisting of four females and four males (23–30 years old) were asked to describe the smell perceived and to record the time and strength after the olfactory tests. All analyses were repeated in triplicate by each assessor under the control environment (temperature: 25 ± 1 °C; relative humidity: 50% ± 10%). A six-point scale ranging from 0 to 5 was applied to assess the aroma intensity, namely 0 (none), 1 (weak), 3 (moderate), and 5 (extreme).

### 2.5. Statistical Analyses

The experiments were performed in triplicate, and the data were expressed as average values. The SPSS statistical package (version 18.0, SPSS Inc., Chicago, IL, USA) was applied for statistical analysis. One-way analysis of variance (ANOVA) was used to compare the results of different samples, and significance was detected when *p* < 0.05.

## 3. Results and Discussion

### 3.1. Total Sugar and Alcohol Contents in Guangdong Hakka Huangjiu

Table 1 exhibits the sugar and alcohol contents of the twenty types of Guangdong Hakka *Huangjiu*. The total sugar content of LPSH, ZJHL-1, and GDSY-1 was less than 100 g/L. Among these, the total sugar content of GDSY-1 was 19.14 g/L (which belonged to the semi-dry type), but the total sugar contents of LPSH and ZJHL-1 were 59.56 g/L and 59.27 g/L (which belonged to the semi-sweet type). In addition, the total sugar contents of the other types of Guangdong Hakka *Huangjiu* were more than 100 g/L (which belonged to the sweet type). In particular, the total sugar content in BZNJ was higher than that in the other types of Guangdong Hakka *Huangjiu*, and its content was up to 299.50 g/L. These differences can be attributed to the addition of the mixed fermentation starter, fermentation time, and operation technology [16]. During the fermentation process of *Huangjiu*, the consumption of sugars is mainly utilized for ethanol fermentation, leading to increases in the alcohol content of *Huangjiu* [1]. In addition, the alcohol contents were regarded as the most important index for assessing *Huangjiu* quality. In this study, a significant difference was found in the alcohol content among the twenty types of Guangdong Hakka *Huangjiu*. Among them, the highest content of alcohol was found in MZSK-1 (35.28 %vol), whereas the lowest content of alcohol was found in GDSY-1 (8.36 %vol). A previous study suggested that a high alcohol content prevents the formation of fruitiness and freshness and masks the main aroma in Chinese rice wine [17]. Furthermore, low-alcohol wine is increasingly preferred by consumers because of the social and health concerns regarding alcohol consumption [18]. These results provided a theoretical basis for the development of Hakka *Huangjiu*.

### 3.2. Main Sugar Content in Guangdong Hakka Huangjiu

Sugar serves as an important carbon source in fermented agri-food products that provides energy in the growth of microorganisms. A previous study showed that the changes in total sugar contents directly reflected the lactic acid bacteria growth to a certain degree [19]. However, some sugars were retained in Chinese rice wine, which had a direct effect on the their quality and aroma [20]. Therefore, the main sugar content in twenty types of Guangdong Hakka *Huangjiu* was detected using HPLC. Glucose and fructose are the most common reducing sugars that widely exist in many agri-food products. As shown in Table 2, the glucose and fructose contents in Guangdong Hakka *Huangjiu* were higher than the contents of sucrose, maltose, and lactose. A previous report exhibited that the concentrations of glucose were significantly altered during the fermentation process of rice wine, which also directly affected the taste substance of the rice wines [21]. Glucose in some beverages could form stable adducts with bisulfite because of the presence of a free carbonyl functional group [22]. In addition, fructose acted as a stronger hydrophobic sugar, and a high fructose concentration effectively elevated the hydrophobicity of Chinese rice wine, which is helpful for promoting the retention of hydrophobic odorants [23]. We also found there was a significant difference in the sucrose, maltose, and lactose contents among the twenty types of Guangdong Hakka *Huangjiu*. Sucrose serves as a crucial raw material that is converted to alcohol and other organic acids during the microbial fermentation process [24]. Maltose consists of two units of glucose that stem mainly from the degradation of starch from raw material [25]. Lactose is a disaccharide milk sugar that is important due to its presence in the many food products, which promotes the mobility of food [26]. Alterations in the main sugar contents of Guangdong Hakka *Huangjiu* lead to the differences in production quality.

### 3.3. Major Organic Acid Contents in Guangdong Hakka Huangjiu

Organic acids, which serve as the crucial ingredients in alcoholic beverages, have a remarkable flavor and cause physiological effects in the body [27]. The organic acid content in Chinese rice wine is associated with the region and is also related to the environment of rice growth. Therefore, the major organic acids in the twenty Guangdong Hakka *Huangjiu* samples were detected using HPLC. As shown in Table 3, the contents of oxalic acid in GDSY-1 were significantly higher than that in the other samples. Oxalic acid was the most common low-molecular-weight organic acid that was produced by fungi and bacteria during the fermentation process [28]. Except for GDSY-1, lactic acid could be detected in the other Guangdong Hakka *Huangjiu* samples. However, the lactic acid content in Guangdong Hakka *Huangjiu* was obviously lower than that in baijiu, probably due to the addition of liquor, which suppressed the growth and reproduction of the lactic acid bacteria [29]. Lactic acid acts as a vital intermediate product that is widely applied in food, cosmetic, and bioplastic production [30]. Interestingly, malic acid was absent in the GDSY-1 and MZHL samples. Malic acid, which plays a crucial role in energy metabolism, can decrease the *Salmonella* content in chicken production and extend product shelf life [31]. In addition, tartaric acid was only present in GDSY-2, ZJHL-1, MZLX-3, MZLX-4, MZSK-1, and MZLWG. Tartaric acid, the most abundant organic acid present in wines, exhibits a remarkable role in maintaining their chemical stability [32]. Tartaric acid is not degraded by microorganisms during the fermentation process of wine, but its concentration is changed by the physiochemical mechanisms [33]. Pyruvic acid is an important keto acid that is mainly produced by *Saccharomyces cerevisiae* during the *Huangjiu* fermentation process. Some pyruvic acid is converted into aldehyde followed by ethanol under anaerobic conditions, which changes the product color and flavor [34]. In the present study, some types of Guangdong Hakka *Huangjiu* had a lower pyruvic acid concentration, especially MZLX-1 and ZZH-2. However, pyruvic acid was not detected in GDSY-1, MZLX-2, MZLX-3, PYHN, or MZSK-1 due to pyruvic acid’s conversion into aldehyde followed by ethanol. During the fermentation process, the carbon source is converted into acetic acid, which is vital to promote cell growth and butyric acid production, and then assimilated to convert acetone, butanol, and ethanol [35]. Citric acid is an intermediate metabolite of the tricarboxylic acid cycle that is mainly obtained from *Aspergillus niger* through submerged fermentation [36]. The citric acid content of *Huangjiu* follows a decreasing order: MZHL > GDSY-1 > LPSH > GDSY-2 > MZLX-1 = BZNJ > MZJX > MZLX-2 > MZKJ = MZLWG = XNHX > MZSK-2 > MZLX-3 = ZJHL-2. Succinic acid is a four-carbon organic acid produced by many microorganisms that can be used in the synthesis of volatile substances such as butanediol, fatty acids, and linear aliphatic esters [37]. Except for MZKJ, succinic acid could be detected in the other Guangdong Hakka *Huangjiu* samples.

### 3.4. The Composition of Amino Acids in Guangdong Hakka Huangjiu

Nitrogenous compounds in raw material, mainly ammonium and amino acids, are utilized by *Saccharomyces cerevisiae* to synthesize proteins, amino acids, nucleotides, and other metabolites. The amino acid contents in Chinese rice wine is influenced by the wine’s aging time, the composition of the raw material, the community structures of the starters, and the fermentation process [38]. Therefore, the composition of amino acids in Guangdong Hakka *Huangjiu* was detected using HPLC. There was a remarkable difference in total amino acid (TAA) contents among the 20 types of Guangdong Hakka *Huangjiu* (Table 4). Among them, the TAA contents in GDSY-1, MZLX-4, and MZLX-3 were more than 400 mg/L, but the TAA contents in PYHN and MZHL were less than 100 mg/L. Amino acids were commonly reported as the precursors of volatile compounds; this is also regarded as another factor that affects wine aroma [39]. Therefore, we speculated that the aromas of GDSY-1, MZLX-4, and MZLX-3 were stronger than those of the other types of Guangdong Hakka *Huangjiu*. In addition, the total amino acid content in Guangdong Hakka *Huangjiu* was significantly lower than that in Hakka *Huangjiu*, probably due to the addition of liquor with a high alcohol content, which suppressed the biochemical processes of fungi and bacteria [39].

At present, amino acids are divided into four types, namely bitter amino acids (BAAs), sweet amino acids (SAAs), umami amino acids (UAAs), and astringent amino acids (AAAs). Thus, their concentrations were further analyzed in the whole Guangdong Hakka *Huangjiu*. The concentrations of taste-determining amino acids in Guangdong Hakka *Huangjiu* followed the trend of BAAs > SAAs > UAAs > AAAs. Among them, BAAs served as the primary amino acids in Guangdong Hakka *Huangjiu*, accounting for more than sixty percent of the total amino acids. A previous study exhibited that the total BAA content in Chinese rice wine was higher than that of other types of amino acids, which was in agreement with our study [40]. In addition, the total SAA contents in GDSY-1 and ZJHL-2 were more than 45 mg/L. The most abundant SAAs in Guangdong Hakka *Huangjiu* were Ser and Thr, which accounted for more than 65% of the total SAAs. Ser promotes cell proliferation and growth and prevents the inflammatory response and oxidative stress [41]. Except for GDSY-1 and MZKJ, the total UAA contents in Guangdong Hakka *Huangjiu* ranged from 14 mg/L to 25 mg/L. Interestingly, Glu was not found in PYHN, probably because *Levilactobacillus* can utilize Glu to produce glutamate decarboxylase in the fermentation process [42]. Astringency, which is the major sensory parameter for assessing the quality of red wine, is strongly associated with the concentrations of Tyr and polyphenols [43]. A significant difference was found in the Tyr concentration among the 20 types of Guangdong Hakka *Huangjiu*. Among them, the Tyr concentration in GDSY-1 was up to 20.98 mg/L, but the Tyr content in MZKJ was 2.31 mg/L.

### 3.5. Volatile Profiles in Guangdong Hakka Huangjiu

The volatile profiles of the twenty types of Guangdong Hakka *Huangjiu* samples were analyzed using HS-SPME coupled with GC-MS. A total of 114 volatiles were separated and identified in these Guangdong Hakka *Huangjiu* samples, including 38 esters, 9 alcohols, 11 aldehydes, 9 ketones, 26 hydrocarbons, 4 phenols, and 17 others. MZSK-1, ZZH-1, and ZZH-2 had the highest abundances of alcohols in comparison with the other types of Guangdong Hakka *Huangjiu* (Figure 1A). Alcohols were reported to provide an important contribution to the presence of the Chinese baijiu aroma, especially fruity and floral characters [44]. Alcohols have an ability to inhibit the “fruitiness” in wines through masking the perception of esters, which can promote the formation of a “metallic” character and a “hotness” sensation [45]. In addition, high abundances of esters were present in MZSK-1. Esters are one of the vital substances of fermentation-associated microbial metabolites that can provide a pleasant fruity aroma in Chinese rice wine. The accumulation of esters results more from the rate of enzymatic synthesis than from hydrolysis reactions and is controlled by the concentrations of two co-substrates and the total ester synthase activity [46]. GDSY-2, ZJHL-1, and MZLX-2 had high abundances of aldehydes and ketones compared with the other Guangdong Hakka *Huangjiu* samples. Aldehydes are common by-products in the processes of food fermentation and degradation that introduce unpleasant flavors and aromas [47]. The increase in the ketone abundance in Chinese rice wine is the most important cause of the citric acid metabolism of lactic acid bacteria strains [48]. In addition, the abundance of alkanes in MZSK-1 was higher than that in the other types of Guangdong Hakka *Huangjiu*. Alkanes are widely regarded as unimportant contributors to fermented food because of their relatively high odor threshold. However, the correlation of aroma substances with the total content of sugars should be further investigated. Furthermore, GDSY-1 and GDSY-2 had a higher abundance of phenols compared with the other Guangdong Hakka *Huangjiu* samples. A hierarchical cluster analysis was used to analyze the overall volatile profiles of the twenty types of Guangdong Hakka *Huangjiu*. As shown in Figure 1B, the twenty types of Guangdong Hakka *Huangjiu* were divided into three groups, namely Group 1 (including GDSY-2, MZLX-2, PYHN, ZZH-1, MZLX-4, GDSY-1, and LPSH), Group 2 (including MZSK-1), and Group 3 (including ZZH-2, BZNJ, ZJHL-1, MZLX-3, MZLX-3, MZLWG, XNHX, MZKJ, MZLX-1, MZJX, MZHL, ZJHL-2, and MZSK-2).

The GC-O technique was carried out to identify the aroma-active ingredients in Guangdong Hakka *Huangjiu*. As shown in Table 5, 18 aroma-active compounds (with odor activity values of more than 1.0) of the Guangdong Hakka *Huangjiu* were detected, including esters (9), aldehydes (4), ketones (2), and alcohols (2). There was a remarkable difference in the aroma-active ingredients and intensities among the twenty types of Guangdong Hakka *Huangjiu*. Most of the aroma-active ingredients identified in the Guangdong Hakka *Huangjiu* had a pleasant aroma of “fruity”. A previous report also exhibited that the esters were the main aroma-active ingredients and mostly had a fruity aroma. The identification of the key aroma ingredients in Guangdong Hakka *Huangjiu* is helpful to consumers in making choices based on their preferences.

## 4. Conclusions

The physicochemical properties and key volatile metabolites in 20 types of Guangdong Hakka *Huangjiu* were compared by using HPLC, GC-MS, and GC-O. A remarkable lower sugar content was found in LPSH, ZJHL-1, and GDSY-1, and a significantly lower alcohol content was found in GDSY-1. There was a significant difference in the organic acids and amino acid composition among the twenty types of Guangdong Hakka *Huangjiu*. In addition, according to the volatile profiles, the twenty types of Guangdong Hakka *Huangjiu* were divided into three groups. However, further studies and investigations such as chemometrics should be performed to develop correlations and relations among these physicochemical properties and key volatile metabolites, and providing a reference description of the indicators within each group would be helpful both for the preparation of standards and for the readers’ understanding. The results obtained in this work offer preliminary insight into the physicochemical properties of Guangdong Hakka *Huangjiu* and provide data support for consumers making choices based on their preferences.

## Figures and Tables

**Figure 1 foods-12-02915-f001:**
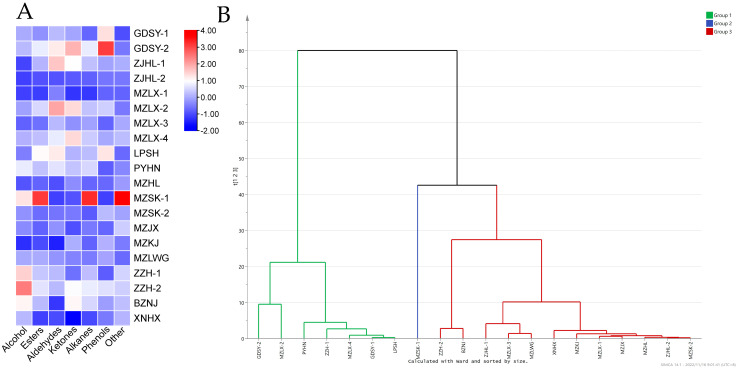
Major volatile profiles in the 20 types of Guangdong Hakka *Huangjiu* samples (**A**). Hierarchical clustering analysis (HCA) of volatile profiles for the 20 types of Guangdong Hakka *Huangjiu* samples (**B**). Group 1 included GDSY-2, MZLX-2, PYHN, ZZH-1, MZLX-4, GDSY-1, and LPSH; Group 2 included MZSK-1; Group 3 included ZZH-2, BZNJ, ZJHL-1, MZLX-3, MZLX-3, MZLWG, XNHX, MZKJ, MZLX-1, MZJX, MZHL, ZJHL-2, and MZSK-2.

**Table 1 foods-12-02915-t001:** Total sugar and alcohol content in twenty types of Guangdong Hakka *Huangjiu*. The total sugar contents were divided into the dry type (less than 15 g/L), semi-dry type (15–40 g/L), semi-sweet type (40–100 g/L), and sweet type (more than 100 g/L).

No.	Sample	Total Sugar (g/L)	Alcohol Content (%vol)	Locations	Types
1	GDSY-1	19.14	8.36	Heyuan City, China	Semi-dry
2	GDSY-2	186.89	10.18	Heyuan City, China	Sweet
3	ZJHL-1	59.27	18.77	Heyuan City, China	Semi-sweet
4	ZJHL-2	121.21	15.54	Heyuan City, China	Sweet
5	MZLX-1	162.98	16.03	Meizhou City, China	Sweet
6	MZLX-2	170.17	11.30	Meizhou City, China	Sweet
7	MZLX-3	142.63	17.56	Meizhou City, China	Sweet
8	MZLX-4	141.25	14.24	Meizhou City, China	Sweet
9	LPSH	59.56	9.97	Meizhou City, China	Semi-sweet
10	PYHN	204.18	18.38	Meizhou City, China	Sweet
11	MZHL	166.01	14.66	Meizhou City, China	Sweet
12	MZSK-1	100.62	35.28	Meizhou City, China	Sweet
13	MZSK-2	168.71	15.64	Meizhou City, China	Sweet
14	MZJX	222.36	12.69	Meizhou City, China	Sweet
15	MZKJ	145.94	15.01	Meizhou City, China	Sweet
16	MZLWG	252.53	11.77	Meizhou City, China	Sweet
17	ZZH-1	243.85	15.07	Meizhou City, China	Sweet
18	ZZH-2	190.34	13.82	Meizhou City, China	Sweet
19	BZNJ	299.50	13.02	Pingyuan City, China	Sweet
20	XNHX	145.94	13.16	Meizhou City, China	Sweet

**Table 2 foods-12-02915-t002:** Main sugar content in twenty types of Guangdong Hakka *Huangjiu*.

Sample	Fructose (mg/L)	Glucose (mg/L)	Sucrose (mg/L)	Maltose (mg/L)	Lactose (mg/L)	Total (mg/L)
GDSY-1	1.20	17.58	0.45	0.25	0.06	19.53
GDSY-2	11.77	129.48	7.58	1.38	4.39	154.61
ZJHL-1	6.20	57.12	0.60	1.09	0.34	65.34
ZJHL-2	9.26	99.26	0.73	1.09	0.87	111.20
MZLX-1	8.52	95.18	1.07	0.73	3.48	108.98
MZLX-2	11.14	123.67	2.93	0.76	0.74	139.25
MZLX-3	9.16	98.76	2.62	1.24	3.42	115.20
MZLX-4	8.45	94.88	1.14	1.16	2.81	108.44
LPSH	11.27	50.03	1.73	1.42	0.91	65.36
PYHN	15.54	162.93	2.93	1.78	1.23	184.40
MZHL	9.96	102.91	1.43	1.62	0.39	116.30
MZSK-1	4.01	84.57	0.77	0.57	1.03	90.94
MZSK-2	12.64	141.33	1.44	2.73	5.67	163.81
MZJX	17.59	204.39	1.24	2.43	0.92	226.56
MZKJ	7.37	72.39	0.27	2.10	1.08	83.21
MZLWG	7.70	185.70	0.83	1.70	2.05	197.98
ZZH-1	14.18	164.75	0.58	1.66	3.30	184.47
ZZH-2	14.27	166.98	11.23	1.39	2.50	196.36
BZNJ	18.26	208.22	0.98	2.91	2.59	232.87
XNHX	17.41	118.62	4.06	4.35	2.32	146.76

**Table 3 foods-12-02915-t003:** The organic acid levels in twenty types of Guangdong Hakka *Huangjiu*.

Sample	Oxalic Acid (mg/mL)	Tartaric Acid (mg/mL)	Pyruvic Acid (mg/mL)	Malic Acid (mg/mL)	Lactic Acid (mg/mL)	Acetic Acid (mg/mL)	Citric Acid (mg/mL)	Succinic Acid (mg/mL)
GDSY-1	12.43	-	-	-	-	-	0.40	1.57
GDSY-2	0.25	0.02	0.05	0.13	8.18	0.92	0.13	4.97
ZJHL-1	0.14	0.23	0.02	0.48	4.91	0.37	0.02	2.70
ZJHL-2	0.31	-	0.04	0.36	11.62	0.27	-	3.48
MZLX-1	0.14	-	0.09	0.17	8.30	0.50	0.09	1.77
MZLX-2	0.63	-	-	2.08	7.72	1.16	0.06	0.57
MZLX-3	0.57	0.54	-	2.21	13.23	0.07	0.02	0.56
MZLX-4	0.39	0.31	0.02	1.22	6.37	0.18	-	0.25
LPSH	1.14	-	0.06	2.56	8.18	0.72	0.23	0.53
PYHN	0.65	-	-	1.72	10.51	0.18	-	1.44
MZHL	0.18	-	0.01	-	6.09	0.33	-	0.09
MZSK-1	0.29	0.23	-	0.87	7.69	-	1.30	0.32
MZSK-2	1.03	-	0.02	2.12	11.76	0.13	0.04	0.79
MZJX	0.69	-	0.04	1.79	10.05	0.23	0.07	0.10
MZKJ	0.37	-	0.01	1.29	4.34	0.21	0.05	-
MZLWG	0.70	0.93	0.03	2.01	8.89	0.48	0.05	0.52
ZZH-1	0.77	-	0.05	1.54	12.17	0.69	-	0.89
ZZH-2	0.97	-	0.08	2.01	13.66	0.42	-	0.22
BZNJ	0.89	-	0.04	1.39	12.35	0.55	0.09	1.14
XNHX	0.66	-	0.01	1.80	12.98	0.49	0.05	0.15

Note: “-” means the content of organic acids did not reach the detection limit.

**Table 4 foods-12-02915-t004:** The amino acid levels in twenty types of Guangdong Hakka *Huangjiu*.

Amino Acid (mg/L)	GDSY-1	GDSY-2	ZJHL-1	ZJHL-2	MZLX-1	MZLX-2	MZLX-3	MZLX-4	LPSH	PYHN	MZHL	MZSK-1	MZSK-2	MZJX	MZKJ	MZLWG	ZZH-1	ZZH-2	BZNJ	XNHX
His	28.46	6.47	7.95	12.16	4.68	5.82	10.32	9.19	10.96	4.06	2.73	8.49	7.39	5.79	177.36	7.50	7.54	7.37	10.72	8.77
Arg	119.51	38.92	42.31	57.62	268.43	241.79	276.45	334.54	47.67	30.30	15.58	44.47	40.37	33.75	10.50	40.93	52.06	44.20	48.09	40.35
Leu	39.34	22.26	20.88	30.56	18.20	14.69	22.54	15.38	20.28	7.732	8.76	22.75	21.45	17.88	6.849	22.9	22.80	23.66	28.23	22.90
Lys	58.52	2.958	16.26	24.41	3.06	1.40	4.27	4.39	6.76	-	-	9.54	4.61	1.04	2.11	5.83	5.09	5.67	1.01	1.02
Val	24.72	13.18	14.35	14.34	16.54	12.05	18.20	8.89	9.86	6.02	5.49	9.77	10.13	11.16	2.81	11.78	9.56	8.28	11.93	10.58
Phe	22.67	15.85	12.68	19.36	8.61	7.19	11.49	7.77	11.34	5.39	6.54	10.87	12.15	13.88	2.69	13.20	14.79	14.56	16.35	14.92
Ile	14.55	7.28	6.66	8.83	8.59	7.255	11.59	4.92	6.55	3.39	3.61	5.90	6.72	5.07	2.06	7.02	6.58	5.88	7.82	6.85
TBAAs	307.77	106.90	121.08	167.27	328.10	290.19	354.85	385.08	113.42	56.88	42.70	111.78	102.82	88.57	204.38	109.16	118.43	109.62	124.14	105.40
Gly	6.66	2.19	3.34	4.26	2.57	2.58	2.38	1.49	2.49	1.25	1.06	2.53	2.13	1.67	1.03	1.98	1.37	1.22	1.61	1.45
Ala	8.23	2.77	5.15	5.71	-	-	1.90	2.04	2.66	-	-	2.54	2.04	3.07	16.37	1.97	2.48	3.41	5.06	5.05
Pro	0.36	-	0.21	0.34	-	-	-	-	-	-	-	-	-	-	-	0.03	0.03	-	-	-
Ser	22.407	6.93	9.08	13.16	6.91	7.96	9.58	7.97	7.34	4.24	3.56	8.08	6.95	5.96	2.83	7.08	6.82	5.88	12.79	8.30
Thr	20.99	11.34	13.00	16.73	8.29	6.75	11.46	6.27	8.99	4.22	4.64	7.17	9.67	9.75	2.31	9.89	10.88	10.29	11.88	12.02
Met	-	-	-	5.13	-	-	3.28	-	3.31	-	-	2.12	-	-	-	4.94	3.43	2.93	4.51	3.72
TSAAs	58.65	23.23	30.78	45.33	17.76	17.29	28.60	17.77	24.80	9.71	9.25	22.44	20.80	20.44	22.54	25.91	25.00	23.73	35.85	30.54
Asp	28.59	17.67	15.23	20.72	16.02	14.99	19.73	11.26	19.94	14.36	9.72	14.18	18.33	13.89	4.72	18.28	15.22	15.78	15.64	13.89
Glu	11.49	3.18	2.67	4.56	2.25	2.31	2.56	2.82	2.80	-	2.32	4.33	3.29	3.73	2.80	4.57	2.70	3.25	7.42	4.94
TUAAs	40.088	20.848	17.903	25.273	18.269	17.307	22.294	14.08	22.732	14.363	12.048	18.516	21.613	17.62	7.524	22.846	17.928	19.029	23.059	18.832
Tyr	20.99	11.34	13.00	16.73	8.29	6.75	11.46	6.27	8.99	4.22	4.64	7.17	9.67	9.75	2.31	9.89	10.88	10.29	11.88	12.02
TAAAs	20.99	11.34	13.00	16.73	8.29	6.75	11.46	6.27	8.99	4.22	4.64	7.17	9.67	9.75	2.31	9.89	10.88	10.29	11.88	12.02
TAAs	427.49	162.32	182.76	254.60	372.41	331.53	417.20	423.20	169.94	85.174	68.633	159.91	154.9	136.38	236.76	167.81	172.23	162.67	194.92	166.79

Note: “-” means the content of amino acids did not reach the detection limit. Ala, alanine; Arg, arginine; Asp, aspartic acid; Glu, glutamic acid; Gly, glycine; His, histidine; Leu, leucine; Lys, lysine; Phe, phenylalanine; Pro, proline; Ser, serine; Thr, threonine; Tyr, tyrosine; Val, valine; TBAAs, total bitter amino acids; TSAAs, total sweet amino acids; TUAAs, total umami amino acids; TAAAs, total astringent amino acids; TAAs, total amino acids.

**Table 5 foods-12-02915-t005:** Volatile profiles in twenty types of Guangdong Hakka *Huangjiu*.

Compounds	GDSY-1	GDSY-2	ZJHL-1	ZJHL-2	MZLX-1	MZLX-2	MZLX-3	MZLX-4	LPSH	PYHN	MZHL	MZSK-1	MZSK-2	MZJX	MZKJ	MZLWG	ZZH-1	ZZH-2	BZNJ	XNHX
2-Ethyl-1-hexanol	-	3.13	-	-	-	-	-	-	-	-	-	-	-	-	-	2.66	-	3.86	2.77	-
phenethyl alcohol	303.39	309.55	47.66	25.05	103.12	334.30	162.72	284.52	186.66	126.71	76.99	31.72	51.26	124.04	30.81	72.98	134.29	240.14	157.91	114.64
ethyl acetate	1.46	3.94	9.24	2.70	1.91	11.62	2.57	2.57	4.53	3.32	2.57	11.76	1.93	0.84	0.53	2.16	5.29	4.05	1.89	0.54
Ethyl butyrate	5.47	6.19	2.21	1.48	1.85	4.51	-	3.87	-	20.41	-	-	-	-	16.89	-	-	-	-	-
Ethyl octanoate	-	-	18.23	13.25	-	-	-	-	13.38	40.18	-	940.14	56.48	5.60	-	15.36	38.06	4.21	8.39	6.54
Ethyl caprate	11.47	8.22	9.73	3.25	1.78	4.28	3.22	-	-	4.89	2.39	962.66	85.71	7.60	2.06	7.62	7.58	5.54	-	4.28
Methyl 2-furanoate	2.27	3.49	0.26	1.09	2.56	7.29	-	1.16	0.61	4.88	-	-	0.67	0.96	-	0.21	0.63	-	-	0.94
Ethyl benzoate	12.08	6.49	5.95	-	4.04	4.38	2.58	7.17	14.82	17.24	3.25	-	4.15	4.39	1.61	2.63	9.69	4.25	-	-
Ethyl phenylacetate	58.62	75.02	47.35	14.96	48.15	226.71	74.75	99.34	94.33	152.71	8.14	-	40.82	7.82	9.81	63.95	81.26	36.75	11.69	-
2-Phenylethyl acetate	-	-	-	-	-	-	6.98	-	-	-	-	-	3.69	9.10	1.70	3.45	-	31.14	58.15	25.61
Diethyl succinate	0.34	0.26	0.13	0.09	0.07	0.30	0.19	0.43	0.54	0.35	0.06	0.07	0.16	0.08	0.05	0.30	0.40	0.22	0.13	0.08
3-Methyl-butyraldehyde	-	-	-	-	-	-	-	-	-	-	-	-	4.32	-	-	2.36	10.29	-	-	-
α- Ethylene-phenylacetaldehyde	134.04	-	-	68.93	89.00	251.94	114.21	227.30	-	-	-	-	-	-	-	-	-	-	-	-
furfural	14.28	34.24	20.44	11.84	12.36	46.35	28.78	22.24	46.54	22.73	12.25	7.52	17.06	23.20	3.32	22.35	24.88	24.62	7.98	8.49
Benzaldehyde	62.11	53.76	122.05	9.78	31.68	48.98	8.72	34.79	8.20	29.66	6.53	11.68	12.28	12.44	5.41	14.37	26.74	29.47	3.02	4.10
2-Octanone	4.23	7.26	5.87	2.95	2.43	6.41	3.86	5.13	4.64	3.12	3.48	3.02	3.04	2.92	3.11	3.83	3.30	5.93	6.23	2.43
2-Nonanone	30.46	45.37	71.75	59.53	18.11	76.21	75.33	64.50	-	69.07	42.44	41.52	29.78	34.55	39.47	15.39	69.78	39.28	30.60	18.49
Acetophenone	27.18	39.25	13.56	3.24	13.40	45.33	8.69	22.50	8.89	38.71	1.89	-	7.51	4.40	1.67	10.80	16.02	27.34	11.07	6.45

Note: “-” means the content of volatile profiles did not reach the detection limit.

## Data Availability

The data used to support the findings of this study can be made available by the corresponding author upon request.

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
