# Peer review of "Comparison Study of the Physicochemical Properties, Amino Acids, and Volatile Metabolites of Guangdong Hakka Huangjiu"

_foods, 2023, doi:10.3390/foods12152915_

Round 1

Reviewer 1 Report

The paper entitled ‘’Comparison study of the physicochemical properties, amino acids, and volatile metabolites of Guangdong Hakka Huangjiu’’ treats an interesting topic aiming to help consumers to make choices based on their preferences.

There are some adjustments that need to be made:

Point 1 (Materials and Methods):

Nothing is mentioned about the 20 types Guangdong Hakka Huangjiu obtained from the local market, in terms of their potential identification / differentiation by ordinary readers. Are the acronyms of the samples related with some specific characteristics of them, such as starter of fermentation, technological steps or others?

Point 2 (Results and discussion):

1. Tables 1 - 5: The SD values for the chemical parameters are missing, although according to point 2.8, ‘’all data were expressed as mean ± SD of three independent experiments (n = 3)’’.

2. Although the discussion part is extensive and detailed, it reported the characteristics of the samples at a certain moment of time. Can be these ones more punctual correlated with the type of the raw materials and ingredients, the technological process or other variables, in order to be helpful both for the preparation of standards, such as the authors mentioned in Introduction part, and for the readers’ understanding?

3. Three groups were identified through Hierarchical cluster analysis (Figure 1B). Understanding the group membership of the samples requires more information than their acronyms.

Author Response

Reviewer #1:

The paper entitled ‘’Comparison study of the physicochemical properties, amino acids, and volatile metabolites of Guangdong Hakka Huangjiu’’ treats an interesting topic aiming to help consumers to make choices based on their preferences.

There are some adjustments that need to be made:

Point 1 (Materials and Methods):

Nothing is mentioned about the 20 types Guangdong Hakka Huangjiu obtained from the local market, in terms of their potential identification / differentiation by ordinary readers. Are the acronyms of the samples related with some specific characteristics of them, such as starter of fermentation, technological steps or others?

Reply: Thank you for your suggestion. More information about 20 types Guangdong Hakka Huangjiu, such as their types and place of origin were provided in Table 1.

Point 2 (Results and discussion):

  1. Tables 1 - 5: The SD values for the chemical parameters are missing, although according to point 2.8, ‘’all data were expressed as mean ± SD of three independent experiments (n = 3)’’.

Reply: Thank you for your careful and patient review. Experiments were performed in triplicate and data were expressed as average values. This expression was modified in the revised manuscript.

  1. Although the discussion part is extensive and detailed, it reported the characteristics of the samples at a certain moment of time. Can be these ones more punctual correlated with the type of the raw materials and ingredients, the technological process or other variables, in order to be helpful both for the preparation of standards, such as the authors mentioned in Introduction part, and for the readers’ understanding?

Reply: Thank you for your careful and patient review. We have determined several main physicochemical properties and volatile metabolites of different kinds of Huangjiu samples. However, these indexes differ from one another. So, further study and investigation such as chemometrics should be performed for developing correlation and relation which are helpful both for the preparation of standards and for the readers’ understanding. It is a subject we are now studying and we are looking forward to submit our following research to this journal as a companion article.

  1. Three groups were identified through Hierarchical cluster analysis (Figure 1B). Understanding the group membership of the samples requires more information than their acronyms.

Reply: Thank you for your valuable suggestion. The groups including different kinds of samples were provided in the figure caption.

Reviewer 2 Report

Plagiarism is 34 %, the authors may be requested to reduce it by 10-15 %

Introudction:

Good enough and addressed the research gaps effectively

Line 39: Qu... stands for ???

Line 184: Text says the data were expressed as means with SD and claimed to have used ANOVA for ranking but in tables no SD values or ranking were depicted. Pl. check and rectify.

Line 200: Pl. check the sentence

clustering is suggested in conclusion section

Author Response

Reviewer #2:

Plagiarism is 34%, the authors may be requested to reduce it by 10-15%.

Reply: Thank you for your valuable suggestion. The repetition rate was reduced by modifying the whole manuscript, especially the Materials and methods section. The repetition rate was reduced to 13% (see the following Figure).

Introduction:

Good enough and addressed the research gaps effectively.

Reply: Thank you for your careful and patient review. Your affirmation has inspired us greatly.

Line 39: Qu... stands for ???

Reply: Qu is a kind of fermentation starter commonly used in rice wine fermentation. “Weat Qu” (Mài QÅ­), “Hong Qu” (Hóng QÅ­) and “Yao Qu” (Yào QÅ­) are the main Qu in Huangjiu preparation.

Line 184: Text says the data were expressed as means with SD and claimed to have used ANOVA for ranking but in tables no SD values or ranking were depicted. Pl. check and rectify.

Reply: Thank you for your careful and patient review. The expression was modified as “Experiments were performed in triplicate and data were expressed as average values.”

Line 200: Pl. check the sentence.

Reply: Sorry for the misunderstanding. This sentence was revised for better understanding. During the fermentation process of Huangjiu, the consumption of sugars was mainly utilized for ethanol fermentation, leading to the increases in alcohol content of Huangjiu

Clustering is suggested in conclusion section.

Reply: Thank you for your careful review and valuable suggestion. It was provided in conclusion section accordingly.

Reviewer 3 Report

The peer-reviewed article presents a large amount of research material that may be of interest to readers. However, for a better perception of the article, it is necessary to finalize the material:

1. In the introduction, it is necessary to give a more detailed description of the technology for making rice wine; Give a description of the microorganisms used in the starter cultures. This will help a wide range of readers to understand what product will be discussed.

2. It is necessary to structure the research results for their better understanding. For example, the authors selected 20 wines of four types (dry, semi-dry, semi-sweet and sweet). Break the 20 images into distinct groups and give a reference description of the indicators within each group.

3. lines 255-266 remove the description of acids, this is not related to wine

4. When describing the results of the determination of aromatic substances, it is desirable to look for a correction with the total content of sugars and alcohol.

5. Conclusions need to be rewritten

Moderate English editing required

Author Response

Reviewer #3:

The peer-reviewed article presents a large amount of research material that may be of interest to readers. However, for a better perception of the article, it is necessary to finalize the material:

  1. In the introduction, it is necessary to give a more detailed description of the technology for making rice wine; Give a description of the microorganisms used in the starter cultures. This will help a wide range of readers to understand what product will be discussed.

Reply: Thank you for your suggestion. Some descriptions were reported in our previous paper and other literatures and a sentence was provided to cite these papers.

  1. It is necessary to structure the research results for their better understanding. For example, the authors selected 20 wines of four types (dry, semi-dry, semi-sweet and sweet). Break the 20 images into distinct groups and give a reference description of the indicators within each group.

Reply: Thank you for your careful and patient review. We have determined several main physicochemical properties and volatile metabolites of different kinds of Huangjiu samples. However, these indexes differ from one another. So, further study and investigation such as chemometrics should be performed for developing correlation and relation which are helpful for giving a reference description of the indicators within each group.

  1. lines 255-266 remove the description of acids, this is not related to wine.

Reply: Thank you for your suggestion. This sentence was deleted accordingly.

  1. When describing the results of the determination of aromatic substances, it is desirable to look for a correction with the total content of sugars and alcohol.

Reply: Thank you for your careful and valuable suggestion. The discussions about the correlation between aroma substances with the total content of sugars and alcohol are provided in the text (marked in red).

  1. Conclusions need to be rewritten.

Reply: Thank you for your careful and valuable suggestion. The conclusion section was rewrited in the revised manuscript.

Round 2

Reviewer 1 Report

Thank you for your answers to the comments that I mentioned! The manuscript was carefully reviewed and improved.

However, there are also some adjustments that need to be made:

1.   It is stated by authors that ANOVA ‘’was used to compare the results of different samples’’ (point 2.5). This statement is not supported as the data are presented.

2.     Figure 1B is unreadable and also, in my opinion, its interpretation could be improved, in order to better understanding of the samples’ clustering, as I have been mentioned previously.